# Improved Discriminability of Severe Lung Injury and Atelectasis in Thoracic Trauma at Low keV Virtual Monoenergetic Images from Photon-Counting Detector CT

**DOI:** 10.3390/diagnostics14192231

**Published:** 2024-10-06

**Authors:** Hanns Leonhard Kaatsch, Maximilian Franz Völlmecke, Benjamin V. Becker, Daniel Dillinger, Laura Kubitscheck, Aliona Wöhler, Sebastian Schaaf, Joel Piechotka, Christof Schreyer, Robert Schwab, Daniel Overhoff, Stephan Waldeck

**Affiliations:** 1Department of Radiology and Neuroradiology, Bundeswehr Central Hospital, 56072 Koblenz, Germany; hannsleonhardkaatsch@bundeswehr.org (H.L.K.);; 2Department of Plastic, Hand and Reconstructive Surgery, Burn Centre, Sarcoma Centre, BG University Hospital Bergmannsheil, 44789 Bochum, Germany; 3Department of Trauma Surgery, Hand and Reconstructive Surgery, Bundeswehr Central Hospital, 56072 Koblenz, Germany; 4Department of Neuroradiology, University Medical Center Mainz, 55131 Mainz, Germany; 5Department of Vascular Surgery and Endovascular Surgery, Bundeswehr Central Hospital, 56072 Koblenz, Germany; 6Bundeswehr Institute of Radiobiology Affiliated to Ulm University, 80937 Munich, Germany; 7Department of General, Visceral and Thoracic Surgery, Bundeswehr Central Hospital, 56072 Koblenz, Germany; 8Department of Radiology and Nuclear Medicine, University Medical Center Mannheim, 68167 Mannheim, Germany

**Keywords:** photon-counting detector CT, virtual monoenergetic imaging, chest trauma, lung injury, atelectasis, emergency radiology

## Abstract

**Objectives**: To evaluate the value of virtual monoenergetic images (VMI) from photon-counting detector CT (PCD-CT) for discriminability of severe lung injury and atelectasis in polytraumatized patients. **Materials & Methods**: Contrast-enhanced PCD-CT examinations of 20 polytraumatized patients with severe thoracic trauma were included in this retrospective study. Spectral PCD-CT data were reconstructed using a noise-optimized virtual monoenergetic imaging (VMI) algorithm with calculated VMIs ranging from 40 to 120 keV at 10 keV increments. Injury-to-atelectasis contrast-to-noise ratio (CNR) was calculated and compared at each energy level based on CT number measurements in severely injured as well as atelectatic lung areas. Three radiologists assessed subjective discriminability, noise perception, and overall image quality. **Results**: CT values for atelectasis decreased as photon energy increased from 40 keV to 120 keV (mean Hounsfield units (HU): 69 at 40 keV; 342 at 120 keV), whereas CT values for severe lung injury remained near-constant from 40 keV to 120 keV (mean HU: 42 at 40 keV; 44 at 120 keV) with significant differences at each keV level (*p* < 0.001). The optimal injury-to-atelectasis CNR was observed at 40 keV in comparison with the remaining energy levels (*p* < 0.001) except for 50 keV (*p* > 0.05). In line with this, VMIs at 40 keV were rated best regarding subjective discriminability. VMIs at 60–70 keV, however, provided the highest subjective observer parameters regarding subjective image noise as well as image quality. **Conclusions**: Discriminability between severely injured and atelectatic lung areas after thoracic trauma can be substantially improved by virtual monoenergetic imaging from PCD-CT with superior contrast and visual discriminability at 40–50 keV.

## 1. Introduction

Pulmonary injuries are common among polytraumatized patients with an incidence up to 40% and are associated with increased mortality [1,2]. The underlying mechanisms of injury can be broadly categorized into blunt, penetrating, ballistic, as well as acceleration-deceleration injuries [3], which may occur in combination. Depending on the severity of the injury, the damage extent can range from diffuse interstitial hemorrhage resulting in pulmonary contusion up to parenchymal tears with hemorrhage in terms of pulmonary laceration or hematoma.

Minor lung contusions usually appear as ground-glass opacification and resolve in the short term [4,5,6]. By contrast, severe lung injuries in terms of vast contusion with consolidation or extensive lung involvement, laceration, and hematoma must be identified early and reliably as they require a different treatment regimen and might cause complications, such as pleural fistula, persisting pneumothorax, pneumonia, abscess formation, pleural empyema, or acute respiratory distress syndrome (ARDS) [7,8,9,10,11,12]. The gold standard of imaging techniques for the assessment of pulmonary injuries after trauma is contrast-enhanced computed tomography (CT) with characteristic image appearances for each of the aforementioned types of pulmonary injuries [13]. However, the transitions between the image appearance of each injury type sometimes appear fluent. This is aggravated by the fact that severe thoracic trauma is often accompanied by pneumo- or hemothoraces, leading to concomitant compression atelectasis of lung tissue. Thus, the differentiation of an increase in lung tissue density in conventional CT due to consolidation in pulmonary injury or due to atelectasis is restricted. In addition, pulmonary injury may additionally be masked by collapsed areas of the lung.

With the introduction of photon-counting detector CT (PCD-CT), spectral information sets are automatically obtained with each examination, which allow the reconstruction of virtual monoenergetic images (VMIs). This technique provides improved contrast of enhancing tissue after contrast medium injection, enabling new diagnostic possibilities, in which a high iodine contrast-to-noise ratio is beneficial. To the best of our knowledge, the use of VMIs in order to distinguish atelectatic from severely injured lung tissue after thoracic trauma has so far not been evaluated. Therefore, our study aimed to assess the value of VMIs derived from PCD-CT scans of severely injured and polytraumatized patients with trauma-induced increases of lung tissue density based on subjective as well as objective image parameters.

## 2. Material and Methods

### 2.1. Ethical Declaration

The present study was approved by the Local Ethics Committee of the Chamber of Physicians Rhineland-Palatinate in Mainz, Germany (number 2022-16314), and conducted in accordance with the Declaration of Helsinki.

### 2.2. Study Population

In this study, all patients admitted to our trauma center from November 2021 up to October 2023 were screened for inclusion and exclusion criteria.

Inclusion criteria were: (a) ICD-10 diagnosis codes S27.0, S27.1, S27.2, S27.31, S27.32, and S27.38 derived from the hospital information system, (b) contrast-enhanced whole-body trauma CT scan performed on a photon-counting CT scanner and (c) simultaneous presence of vast pulmonary contusion/laceration/hematoma and atelectasis. Exclusion criteria were severe artifact burden due to breathing motion (*n* = 2), photon starvation (*n* = 2), or beam hardening affecting the visibility and reliable measurability of pulmonary injury or atelectasis. Figure 1 provides an overview of the patient-selection process.

In order to assess trauma severity of the study cohort, the Abbreviated Injury Scale (AIS) score of the thorax as well as the Injury Severity Score (ISS) were calculated. Detailed information is provided in Table 1.

### 2.3. Data Acquisition

After primary survey a whole-body trauma CT scan was performed on a dual-source photon-counting CT scanner (NAEOTOM Alpha, Siemens Healthineers, Forchheim, Germany), consisting of an unenhanced head CT, a CT angiography of head and neck, as well as a contrast-enhanced thoracoabominal CT. Acquisition times were 4.4 s for head CT, 3.2 s for CT angiography of head and neck, and 3.2 s for thoracoabdominal CT. Automatic tube current modulation was enabled (CARE Dose4D, Siemens Healthineers, Forchheim, Germany) and the patients were scanned in cranio-caudal direction with arm position at the patient’s side except for one patient with arms raised above the head. Scan parameters for the thoracoabdominal CT were as follows: Scan mode: Quantum plus, tube voltage: 2 × 120 kV, collimation: 2 × 144 × 0.4 mm, pitch: 0.8, rotation time: 0.25 s, software version VA40. A split-bolus contrast injection protocol was applied with intravenous injection of a total of 120 mL contrast agent (Xenetix 350, 76.78 g/100 mL Iobitridol, Guerbet, Roissy, France) split into an initial bolus of 60 mL followed by a saline flush of 50 mL as well as a time delay of 15 s and a second bolus of 60 mL followed by a saline flush of 50 mL with a flow rate of 4 mL/s each using a power injector (CT Motion XD8000 (Ulrich Medical, Ulm, Germany)). Virtual monoenergetic images (VMI) of 40–120 keV with a 10 keV increment were reconstructed from spectral post processing (SPP) datasets applying the VMI+ application (Siemens Healthineers, Erlangen, Germany) with a soft tissue convolution kernel (Qr40), quantum iterative reconstruction (QIR) level 3, image matrix of 512 × 512, and a slice thickness and increment of 1.0/0.7 in the axial plane.

The effective dose was calculated using Radimetrics 3.4.2 (Bayer AG, Leverkusen, Germany) according to ICRP 103 using individual Monte Carlo simulation with phantoms individually adapted to patient size, scan range, and mAs modulation.

### 2.4. Objective and Subjective Image Parameters

Both objective as well as subjective image analysis were performed on a dedicated workstation (Syngo.via, VB60A, Siemens Healthineers, Forchheim, Germany). For each scan two circular regions of interest (ROIs) of 1 cm^2^ were placed in consecutive images of atelectasis as well as extensively injured lung parenchyma (vast contusion with consolidation, laceration, and hematoma) with exclusion of larger pulmonary vessels and bronchi at 70 keV. The placement of ROIs was performed in consensus by two radiologists with 9 and 13 years in trauma imaging. Mean and standard deviation of CT numbers (Hounsfield units (HU)) were recorded, and measurements of two ROIs were averaged for all aforementioned energy levels as the ROIs remained consistent due to an automated reconstructing process. Following Konietzke et al. the contrast-to-noise ratio (CNR) between two tissues (1 and 2) was calculated as follows with S representing averaged HU and σ representing the averaged standard deviation [14]:(1)$$CNR=\frac{\left|S_{1}-S_{2}\right|}{\sqrt{0.5(\sigma_{1}^{2}+\sigma_{2}^{2})}}$$

Subjective image quality was evaluated by three radiologists with 9 years, 12 years, and 13 years extensive experience in trauma imaging independently in a randomized order, blinded fashion, and under constant conditions. The readers were free to scroll through the entire stack of 3D images and adjust window settings. Based on a 5-point Likert scale, general image quality (1 = very poor, 2 = poor, 3 = acceptable, 4 = good, 5 = optimal), image noise (1 = major noise, 2 = more than average noise, 3 = average noise, 4 = minor noise, 5 = insignificant noise), and discriminability of lung injury and atelectasis were assessed (1 = no differentiation, 2 = poor differentiation, 3 = mostly discriminable, 4 = good differentiation, 5 = excellent differentiation).

### 2.5. Data Analysis

All analyses were performed using R Statistical Software [15] (R version 4.0.4; RStudio version 1.4). For data visualization ggplot2 and Likert package [16,17] were used. A Q-Q plot was used to assess for normal distribution. After exclusion of extreme outliers, a two-repeated measures analysis of variance (ANOVA) was performed to evaluate whether measured HU values of atelectasis and severe lung injury differed over keV levels. The resulting *p*-values were corrected by using the Bonferroni multiple testing correction method. A paired *t*-test was used for post-hoc testing at all keV levels, again considering the multiple testing correction by applying the Bonferroni method. Regarding calculated CNR values, after assessing normal distribution by Q-Q plot and exclusion of extreme outliers, a one-way repeated measures ANOVA was used to test for significant difference of CNR over all keV levels. Paired *t*-test was used to assess whether CNR differed among each keV level. The resulting *p*-values of both one-way measures ANOVA and paired *t*-test were corrected using the Bonferroni multiple testing correction method. Intraclass correlation coefficient (ICC) was computed for absolute agreement in a two-way random effects approach to measure interrater agreement. Agreement was interpreted as follows: >0.9 excellent, 0.9–0.76: good, 0.75–0.51: moderate, <0.5: poor.

## 3. Results

Example images of an identical patient depict trauma-related pulmonary opacifications reconstructed in a standard lung tissue kernel (Figure 2) and in virtual monoenergetic images ranging from 40 keV up to 120 keV applying a standard soft tissue kernel (Figure 3) of contrast-enhanced photon-counting detector CT.

### 3.1. Patient Demographics and Dose Exposure

The total number of patients included was 20. The mean patient age was 53.9 ± 17.4 years (range: 20–81 years) with 19 male and 1 female patient. Mean DLP of contrast-enhanced thoracoabdominal CT from whole-body trauma CT was 897.84 ± 375.97 mGy∗cm, and mean CTDIvol was 12.36 ± 3.46 mGy. Calculated mean effective dose was 12.02 ± 3.98 mSv.

### 3.2. Quantitative Image Analysis

Density measurements are depicted in Figure 4. Mean HU values in atelectasis gradually decreased from 342 ± 97 HU at 40 keV to 69 ± 15 HU at 120 keV. Mean HU values in severe lung injury were measured lowest at 40 keV with 42 ± 49 HU with a minimal increase to 44 ± 17 HU at 80 keV and a stagnation with further increasing keV levels measuring 44 ± 22 HU at 120 keV. Low-energy keV levels were characterized by a larger variance of HU values compared to high-energy keV levels. The mean HU values between lung injury and atelectasis were significantly different at all keV levels (*p* < 0.001). In addition, the contrast-to-noise-ratio (CNR) between severely injured lung tissue and atelectasis was calculated for all keV-levels (Figure 5). The highest CNR was achieved at 40 keV (3.97) with a continuous decrease with increasing energy levels down to a minimum value of 1.21 at 120 keV. CNR reached no significant differences between 40 and 50 keV as well as 110 and 120 keV (*p* > 0.05), all other pairings showed statistically significant differences (*p* < 0.05).

### 3.3. Qualitative Image Analysis

Detailed results are provided in Table 2. The highest ratings for subjective discriminability of atelectasis and lung injury by all raters were achieved at 40 and 50 keV (4.66 ± 0.69 and 4.44 ± 0.56, respectively) with a continuous decrease of rating scores with increasing energy level down to 1.18 ± 0.43 at 120 keV. Good to excellent differentiability between severe lung injury and atelectasis was assessed in 92% and 95% of ratings at 40 and 50 keV, respectively, whereas the discriminability from 90 keV upwards decreased rapidly and was mostly rated as poor or indistinguishable (Appendix A). In regard to subjective image noise, the highest ratings were achieved at 70 and 80 keV (4.14 ± 0.39 and 4.14 ± 0.47, respectively) with a range of predominantly minor or non-relevant noise burden in VMIs from 60 up to 90 keV and worst ratings in the respective margin energy levels (Appendix A). These results were mirrored by the ratings of overall subjective image quality. Leading optimal as well as good subjective image quality ratings were achieved at 60 to 90 keV with the highest scores at 70 keV (4.34 ± 0.48) and the lowest scores at 120 keV (3.24 ± 0.82; Appendix A). Adding up the scores from all readers and all evaluation categories to an overall score resulted in the highest overall score for 60 keV closely followed by 70 keV (Table 2; Figure 6). Good agreement was found between the three raters for the overall subjective ratings (intraclass correlation coefficient: 0.75).

## 4. Discussion

This study evaluated virtual monoenergetic imaging for improved differentiability of severe lung injury from atelectasis in contrast-enhanced PCD-CT scans of polytraumatized patients with acute thoracic trauma. Based on our quantitative and qualitative image analysis, VMIs at 40–50 keV provided the highest injury-to-atelectasis contrast, while VMIs at 60–70 keV provide the best compromise between sufficient discriminability, the lowest level of subjective noise perception, and additional consideration of subjective overall image quality.

The diagnostic advantage of VMI reconstructions at low keV levels derived from contrast-enhanced spectral CT datasets, on which this study is based, is due to the approximation to the K-edge of iodine (33 keV). By virtually lowering the energy level, CT attenuation of iodinated tissue, such as properly perfused atelectatic lung tissue, is increased [18,19,20]. As a result, non-perfused or perfusion-restricted tissue, such as severely injured lung tissue, becomes visually distinguishable due to a comparably reduced or missing radiodensity. The traditional reconstruction of VMIs from dual-energy CT (DECT) datasets were initially marked by increased image noise at low keV levels, ultimately hampering image quality and a decline of CNR at low keV values [21,22,23]. This limitation was addressed by the introduction of a sophisticated virtual monoenergetic reconstruction algorithm—the VMI+ application (Siemens Healthineers, Erlangen, Germany) [24]. By combining increased iodine attenuation from low energies with superior noise properties from medium energies into composite image stacks, a reduction of image noise concomitant with an improved CNR of iodine compared to prior VMI reconstructions was achieved, especially at low keV levels [24,25,26]. Consequently, the application of this reconstruction algorithm in our study resulted in an optimal CNR as well as superior subjective discriminability between severe lung injury and atelectasis at 40 keV with a continuous decline of quantitative as well as qualitative parameters as energy levels increased. Despite improved handling of image noise burden at low keV levels of VMI+ reconstructions, however, comparably minor image noise with impact on image quality still remained. This is reflected by larger variance of HU values derived from low keV levels and the highest subjective image quality as well as subjective image noise ratings by our raters at midrange keV levels between 70–80 keV, which is congruent with a previous study assessing observer preference in VMI+ images of DECT angiography [26].

To our knowledge, this is the first study investigating the usability of VMIs to differentiate between severe lung injury and atelectasis after thoracic trauma in polytraumatized patients. Nevertheless, VMIs derived from DECT scanners have already been used successfully for the diagnostics of other severe organ injuries associated with structural damage and impaired perfusion. In line with our results, the optimal injury-to-parenchyma contrast of liver and splenic laceration as well as the highest CNR for the assessment of pancreatic laceration in abdominal trauma patients have been reported for VMIs at 40 keV [27,28]. In the context of non-traumatic causes for increases in lung parenchyma density, quantitative differentiation of pneumonia and atelectasis based on contrast-enhanced chest CT has previously been investigated. Both spectral CT parameters in terms of iodine concentration and effective atomic number as well as CT number measurements successfully proved their respective diagnostic use for this purpose with no added value of spectral information compared to density measurements based on X-ray attenuation in conventional CT images [14,29].

Severe artifact burden led to the exclusion of four patients in our study. Potential strategies to counter the artifact burden do exist; unfortunately, they cannot always be implemented in clinical trauma care. In regard to photon starvation, which led to the exclusion of two patients in our study, elevated positioning of the arms represents the most effective solution. Polytraumatized patients, however, might show injuries that prevent arm elevation or require the inclusion of shoulder girdle and arms in the field of view. In addition, arm positioning above the head would usually require an interruption after head CT and CT angiography of head and neck leading to an extended scan time, which would have to be weighed against the actual diagnostic benefit in a time-critical trauma setting. In clinical practice, reimaging after stabilization with elevation of the arms may prove useful in this context. Severe obesity causing artifacts poses a severe diagnostic challenge. Despite maximal exposure factors or automated tube current modulation across different areas of the body with varying thickness, considerable artifacts might nevertheless remain, or even non-diagnostic scans might be caused due to obesity. Technical limits seem to have been reached for the time being in this context. In regard to breathing motion artifacts, which led to exclusion of two patients in our study, awake and compliant patients are prompted to hold their breath before the thoracoabdominal scan of whole-body trauma CT. Polytraumatized patients, however, are often clouded in their consciousness, sedated, or intubated, hampering active cooperation during the scan. Respiratory triggering or endexpiratory breathing arrest controlled by a ventilator machine may help to overcome severe artifacts caused by breathing. Lastly, beam hardening artifacts due to radiopaque metallic foreign objects located in- and outside of the patient may impair image quality. Next to removal of non-essential radiopaque material prior to the CT scan or placement outside of the field of view, the use of modern metal artifact reduction techniques may prove useful in this context.

Despite adequate early treatment with insertion of a chest tube and re-expansion of the lung, persistent parenchymal lung injury after severe thoracic trauma occurs in up to 30% of patients with corresponding late complications [7,8,30,31]. Extensive lung injury also carries a risk for pulmonary dysfunction in terms of development of ARDS in the further clinical course [12,32]. Improved detectability and conclusive differentiation of severe lung injury from atelectasis at an early stage of clinical treatment is therefore essential for clinical decision-making. With this ability provided by VMIs reconstructed from spectral datasets of a PCD-CT, patients at a high risk can be identified more precisely, which leads to a distinguished decision pattern regarding admission to intensive care units for adequate ventilation therapy. In this respect, our study underlines the value of inherent spectral data of PCD-CT scans in emergency diagnostics, ultimately improving patient outcome and reducing morbidity as well as mortality. Concerning adequate image reconstruction in whole-body trauma PCD-CT, spectral reconstructions using a soft kernel (Qr40) proved to be useful in addition to a sharp standard kernel (Bl60) for the lung in our approach.

In regard to the determination of the extent of severe lung injury with predictive value, our findings may help to improve existing methods for quantification of pulmonary injury [2,10,11,12]. Next to enhanced manual quantification, VMIs with increased contrast of tissue densities could also establish a basis for automated quantification or color-coded 3D visualization of severe lung injury. This may pave the way for further studies focusing on the application of artificial intelligence for computer-aided detection or on the prediction of patient outcome, e.g., the overall length of the hospital stay, the need for intensive care treatment, as well as the development of ARDS in correlation to the extent of lung damage.

The following limitations must be acknowledged. First of all, our study is lacking a validated reference for either lung injury or atelectasis since no histological correlation was performed, which harbors the risk of misinterpreting of the results. Our classification to either entity was based on image morphology, history of acute thoracic trauma, and density measurements, which we believe, however, allows clear differentiation of severe lung injury, atelectasis, and other differential diagnoses. This is supported by the fact that comparable density values for atelectasis were measured in previous studies when compared to pneumonia, which also makes this entity unlikely as a differential diagnosis for lung opacification [14,29]. Furthermore, an interchange in the classification of lung injury and atelectasis is unlikely in view of the statistically highly significant differences in density values for each keV level with pathophysiologically logical explainability. Minor areas of severe lung injury might nevertheless be masked by collapsed, non-aerated lung regions, which cannot be identified until restored proper ventilation of the lung and reimaging in the further treatment course. Artifact burden in whole-body trauma CT is commonly encountered with a potential influence on diagnostic accessibility as well as reliable quantitative density measurements. In regard to this study, severe image artifacts resulted in the exclusion of four patients, which in clinical practice would require reimaging in the further clinical course. In further studies it would be interesting to analyze to what extent modern artifact suppression software could positively influence the overall artifact burden in whole-body trauma CT [33,34]. Moreover, it should be noted that this is a single-center study including a rather small sample size and that our findings cannot be generalized for all CT scanners from different vendors, as we used a dual-source PCD-CT with a vendor-specific virtual monoenergetic reconstruction algorithm. In regard to the aforementioned advantages of this reconstruction technique, it should be taken into account that the extent of image noise may vary when using other devices and reconstruction techniques. Finally, no comparison between virtual monoenergetic reconstructions and conventional polychromatic CT was performed. Considering VMIs at approximately 75 keV as equivalent to 120 kVp conventional CT [18], several studies focusing on image quality of both reconstructions revealed superior results for conventional-equivalent VMIs [35,36,37], which in our view makes an additional comparison of both modalities not mandatory.

In conclusion, we demonstrated that noise-optimized, low-energy VMIs derived from photon-counting whole-body CT enhanced the differentiability of severely injured lung parenchyma from atelectasis in polytraumatized patients with acute thoracic trauma. This enables a reliable detection as well as more precise assessment of the extent of severe lung damage in emergency imaging and supports clinicians with the initiation of adequate therapy and the anticipation of possible trauma-related complications in the further clinical course.

## Figures and Tables

**Figure 1 diagnostics-14-02231-f001:**
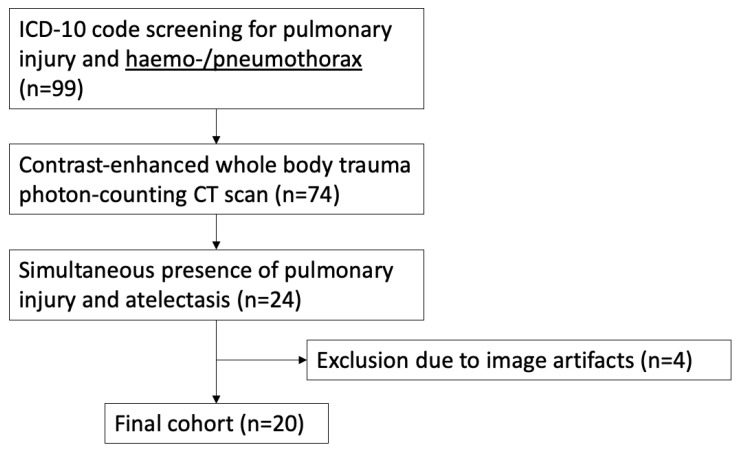
Flow chart for patient selection with regard to inclusion and exclusion criteria.

**Figure 2 diagnostics-14-02231-f002:**
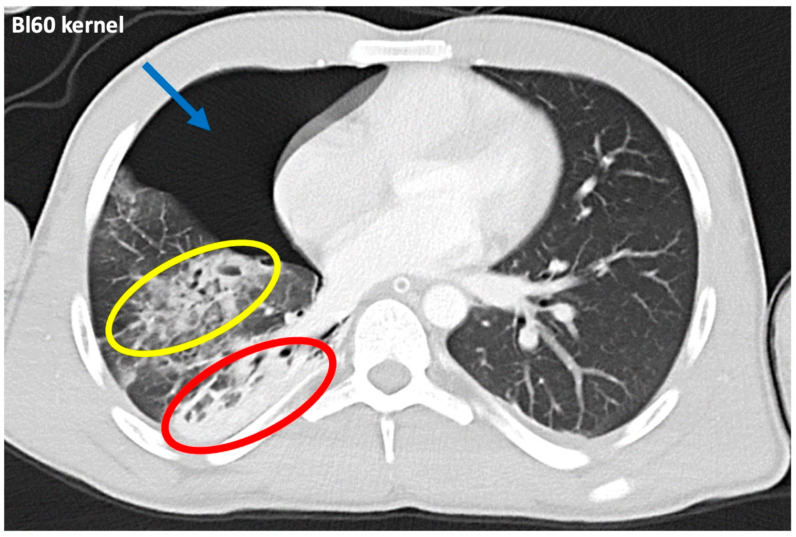
Example images of two areas of pulmonary opacification (red and yellow oval-shaped markers) in the partially collapsed upper lobe and lower lobe of the right lung after thoracic trauma concomitant with pneumothorax (blue arrow). The yellow marker circles pulmonary laceration, and the red marker circles atelectatic lung tissue. Images were reconstructed in a standard lung tissue kernel (Bl60) of contrast-enhanced photon-counting detector CT.

**Figure 3 diagnostics-14-02231-f003:**
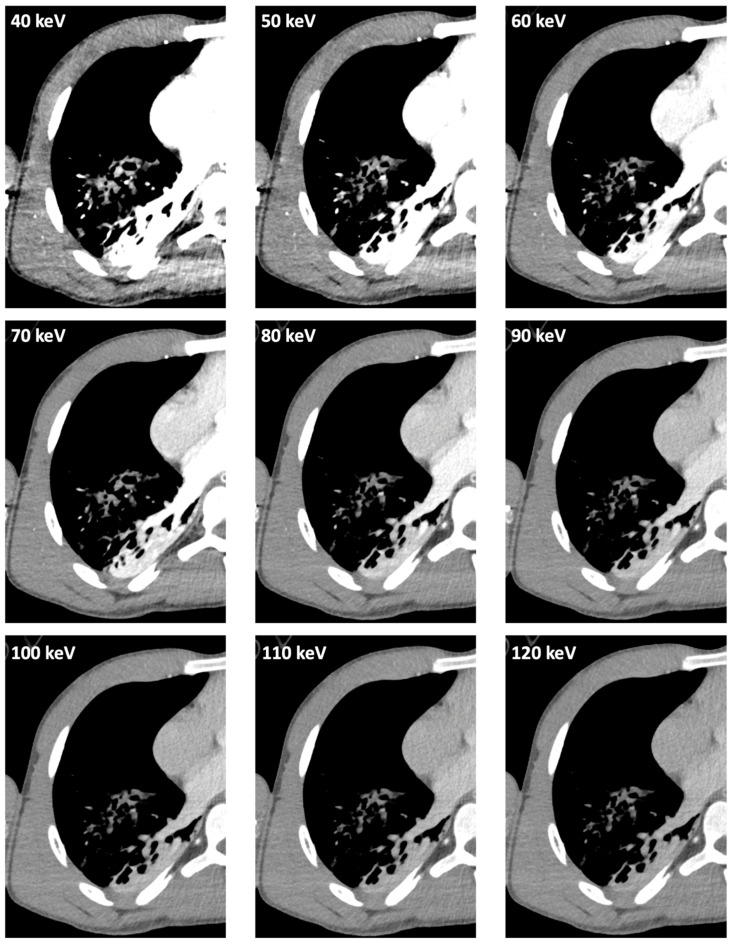
Example images of the identical patient as in Figure 2 reconstructed in virtual monoenergetic images of contrast-enhanced photon-counting detector CT ranging from 40 keV up to 120 keV applying a standard soft tissue kernel (Qr40). The example images of each keV-level depict that low-energy VMIs provide improved visual discriminability between both areas of pulmonary opacification (see Figure 2) accompanied by an increase of image noise. In contrast to this, high keV levels are characterized by a distinctly poorer discriminability.

**Figure 4 diagnostics-14-02231-f004:**
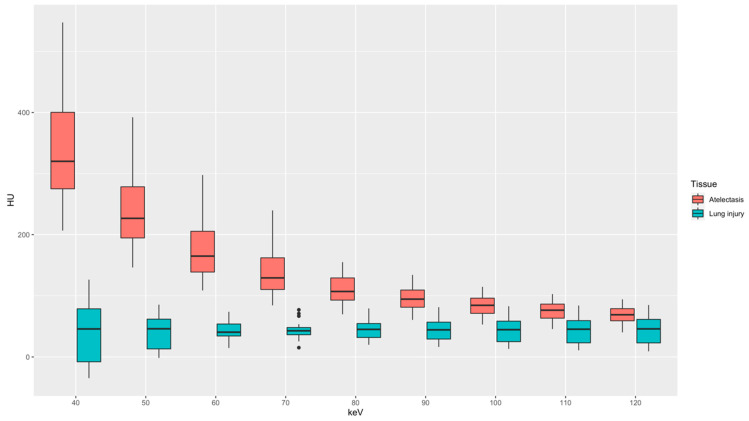
Boxplot displaying mean HU values of atelectasis and lung injury. Boxes display the interquartile range with the median as a horizontal line, whiskers for minimum and maximum, and outliers as dots. Pairwise comparison of mean HU values differed significantly at all keV levels (*p* < 0.001).

**Figure 5 diagnostics-14-02231-f005:**
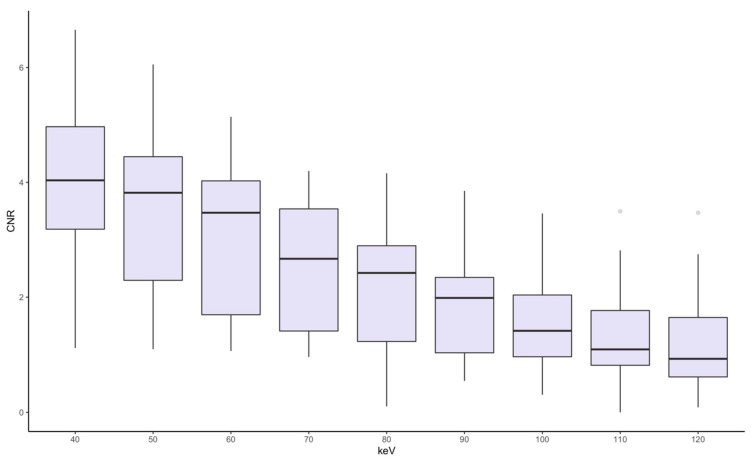
Boxplot displaying calculated CNR between atelectasis and lung injury for all keV levels. Boxes display the interquartile range with the median as a horizontal line, whiskers for minimum and maximum, and outliers as dots. Pairwise comparison of CNR values differed significantly at all keV levels (*p* < 0.05) except for 40–50 and 110–120 keV.

**Figure 6 diagnostics-14-02231-f006:**
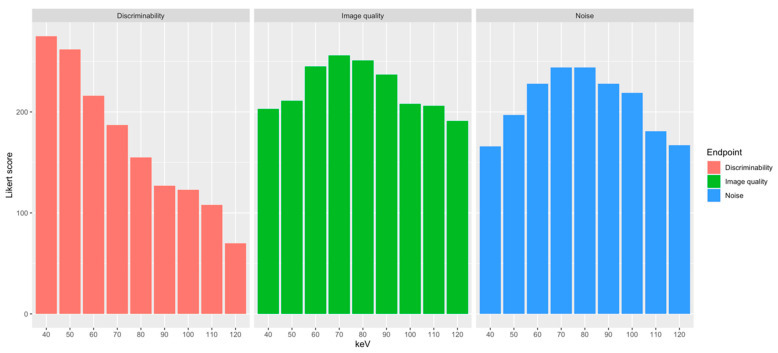
Bar charts of the overall scores by the three raters for each energy level and assessment category in subjective image analysis.

**Table 1 diagnostics-14-02231-t001:** Injury Severity Score (ISS) and Abbreviated Injury Scale (AIS) score of the chest data as well as a detailed list of injury patterns.

ISS		AIS Chest	
ISS-category	*n*=	AIS-category	*n*=
74–50 (critical)	3	5—critical	5
49–25 (severe)	9	4—severe	4
24–16 (serious)	4	3—serious	10
15–05 (moderate)	4	2—moderate	1
**Type of Chest Injury**		**Frequency**	
Pulmonary contusion		19	
Pulmonary laceration		16	
Haematothorax		14	
Pneumothorax		20	
Mediastinal hematoma		2	
Bronchial rupture		1	
Diaphragmatic rupture		1	
Pneumomediastinum		4	
Soft tissue emphysema		12	
Singular rib fracture		1	
Serial rib fracture		18	
Thoracic spine injury		7	
Bony shoulder girdle injury		12	

**Table 2 diagnostics-14-02231-t002:** Subjective image quality scores of all readers for each keV level of VMIs in respect to discriminability of lung injury and atelectasis, image noise, and image quality. Respective values are reported as means and standard deviation (SD). In addition, overall scores of all readers for each energy level are reported.

	Mean ± SD			
Energy Level	Discriminability	Noise	Image Quality	Overall Score
40 keV	4.66 ± 0.69	2.81 ± 0.6	3.44 ± 0.6	644
50 keV	4.44 ± 0.56	3.34 ± 0.61	3.58 ± 0.53	670
60 keV	3.66 ± 0.69	3.86 ± 0.47	4.15 ± 0.48	689
70 keV	3.17 ± 0.77	4.14 ± 0.39	4.34 ± 0.48	687
80 keV	2.63 ± 0.79	4.14 ± 0.47	4.25 ± 0.51	650
90 keV	2.15 ± 0.66	3.86 ± 0.73	4.02 ± 0.68	592
100 keV	2.08 ± 0.92	3.71 ± 0.85	3.53 ± 0.8	550
110 keV	1.83 ± 0.91	3.07 ± 1.27	3.49 ± 0.8	495
120 keV	1.18 ± 0.43	2.83 ± 1.29	3.24 ± 0.82	428

## Data Availability

The data of the current research can be requested from the corresponding author.

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
