# Peer review of "Improved Discriminability of Severe Lung Injury and Atelectasis in Thoracic Trauma at Low keV Virtual Monoenergetic Images from Photon-Counting Detector CT"

_diagnostics, 2024, doi:10.3390/diagnostics14192231_

Round 1

Reviewer 1 Report

Comments and Suggestions for Authors

Thank you for submitting your manuscript “Improved discriminability of severe lung injury and atelectasis in thoracic trauma at low keV virtual monoenergetic images from Photon-Counting detector CT” for consideration. I was pleased to receive it as a reviewer.

Your work has several strengths, including:

-          A novel application of virtual monoenergetic imaging.

-          Rigorous quantitative and qualitative image analysis.

-          Clear presentation of results with helpful figures and tables.

-          Discussion of potential clinical implications for improving trauma care.

To further strengthen the manuscript, I have the following suggestions:

1.      The small sample size (n=20) is a significant limitation. Have you considered performing a power analysis to determine if this sample size is sufficient to support your conclusions?

2.      The study lacks a validated reference standard for lung injury versus atelectasis. Could you elaborate on how this limitation may impact the interpretation of your results?

3.      The study focuses on thoracic trauma, but whole-body trauma CT scans were performed. Have you considered analysing the potential benefits of VMI for detecting injuries in other body regions from the same dataset?

4.      Have you considered performing ROC analysis to determine optimal keV levels for discriminating injury from atelectasis?

5.      In Figure 4, the HU values for lung injury appear to have larger variance at lower keV levels. Can you comment on the potential reasons for this increased variability?

6.      Could you provide more details on the types and frequencies of artifacts encountered, and discuss potential strategies to mitigate these in future studies?

7.      You mention potential applications for automated quantification and AI-based detection. Could you expand on specific next steps or ongoing work in this direction?

8.      Given that your results are specific to one PCD-CT scanner model, how generalizable do you think these findings are to other spectral CT technologies?

9.      The potential for VMI to improve existing methods of quantifying pulmonary injury is interesting. Do you have any preliminary data on correlations between VMI-based injury assessments and clinical outcomes?

I look forward to seeing the revised manuscript.

Author Response

Reviewer 1

Thank you for submitting your manuscript “Improved discriminability of severe lung injury and atelectasis in thoracic trauma at low keV virtual monoenergetic images from Photon-Counting detector CT” for consideration. I was pleased to receive it as a reviewer.

Your work has several strengths, including:

-          A novel application of virtual monoenergetic imaging.

-          Rigorous quantitative and qualitative image analysis.

-          Clear presentation of results with helpful figures and tables.

-          Discussion of potential clinical implications for improving trauma care.

To further strengthen the manuscript, I have the following suggestions:

We are grateful for the fair, constructive and helpful comments raised by Reviewer 1. Please find our point-by-point responses to your remarks in the following.

  1. The small sample size (n=20) is a significant limitation. Have you considered performing a power analysis to determine if this sample size is sufficient to support your conclusions? 

As stated above our study represents a novel application of VMI consisting of a descriptive retrospective evaluation of the existing data in terms of a proof-of-concept / pilot study. Therefore, we performed a hypotheses-generating approach, on the basis of which an effect size could be estimated for validation in the next step. Since the effect size is not yet determined and there is no comparable data, a power analysis is not applicable in our view.

  1. The study lacks a validated reference standard for lung injury versus atelectasis. Could you elaborate on how this limitation may impact the interpretation of your results?

Due to the lack of a reference standard, it is theoretically conceivable that the two entities could have been interchanged or that there could have been other reasons for lung opacification. Both would significantly affect the interpretation of our results and might lead to a false assumption. In our opinion, however, the probability of this is extremely low, especially when the results of our density measurements are taken into account. These show a statistically highly significant differentiability of both entities for each energy level. In particular, constant mean HU values of approximately 42-44 HU for pulmonary injury were measured across all keV levels, which support our assumption as injured, non-perfused and haemorrhaged lung tissue. Atelectasis, on the other hand, is pathophysiologically due to lung collapse with maintained regular perfusion, which is mirrored by increasing HU values obtained by our measurements as energy levels decrease due to iodine-induced enhanced absorption of photons. This is supported by the fact, that comparable density values for atelectasis were measured in previous studies [1, 2] when compared to pneumonia, which also makes this entity unlikely as a differential diagnosis of lung opacification in our study. Considering all of this, we are convinced that our approach allows sufficient differentiability of lung injury and atelectasis from other reasons for lung opacification in temporally-related acute trauma.

We thank Reviewer 1 for this remark and adjusted the manuscript. It now reads:

“The following limitations must be acknowledged. First of all, our study is lacking a validated reference for either lung injury or atelectasis since no histological correlation was performed, which harbors the risk of misinterpreting the results. Our classification to either entity was based on image morphology, history of acute thoracic trauma and density measurements, which we believe, however, allows clear differentiation of severe lung injury, atelectasis and other differential diagnoses. This is supported by the fact, that comparable density values for atelectasis were measured in previous studies when compared to pneumonia, which also makes this entity unlikely as a differential diagnosis for lung opacification [1, 2]. Furthermore, an interchange in the classification of lung injury and atelectasis is unlikely in view of the statistically highly significant differences in density values for each keV level with pathophysiologically logical explainability.“

  1. The study focuses on thoracic trauma, but whole-body trauma CT scans were performed. Have you considered analysing the potential benefits of VMI for detecting injuries in other body regions from the same dataset?

In our practical experience, low-energy VMIs and iodine maps proved to be particularly useful for fast detection of smaller sources of bleeding, such as mesenteric vessel tears and intercostal artery bleeding, or abdominal organ lacerations. However, the total number of patient cases with concomitant either of the aforementioned injury derived from this dataset is not sufficient. As a consequence, we plan to collect more patients and address these aspects by further studies in the near future.

  1. Have you considered performing ROC analysis to determine optimal keV levels for discriminating injury from atelectasis?

We have not considered to perform further statistical analysis, since performed statistical analysis for quantitative and qualitative image assessment in this descriptive proof-of-concept approach unrestrictedly support the conclusions drawn from this by us.

  1. In Figure 4, the HU values for lung injury appear to have larger variance at lower keV levels. Can you comment on the potential reasons for this increased variability?

In our opinion, this larger variance of HU values for lung injury but also atelectasis at low keV levels is mainly due to noise affliction of low-energy VMIs. We thank Reviewer 1 for pointing this out and added an additional sentence to our Results and Discussion section:

“Low-energy keV levels were characterized by a larger variance of HU values compared to high-energy keV levels.“

“Despite improved handling of image noise burden at low keV levels of VMI+ reconstructions, however, comparably minor image noise with impact on image quality still remained. This is reflected by larger variance of HU values derived from low keV levels and highest subjective image quality as well as subjective image noise ratings by our raters at midrange keV levels between 70-80 keV, which is congruent with a previous study assessing observer preference in VMI+ images of DECT angiography [3].”

  1. Could you provide more details on the types and frequencies of artifacts encountered, and discuss potential strategies to mitigate these in future studies?

In total, four patients were excluded from our analysis. Two patients were excluded due to streak artifacts caused by photon starvation in non-elevated positioning of the arms aggravated by concomitant obesity. The other two patients were excluded due to breathing motion artifacts. Further artifacts encountered during our analysis were beam hardening artifacts due to radiopaque metallic foreign objects located in- and outside of the patient, though the extent of these artifacts did not lead to exclusion in our study.

Potential strategies to counter the artifact burden do exist, unfortunately, they cannot always be implemented in clinical trauma care. In regard to photon starvation, which led to exclusion of two patients in our study, elevated positioning of the arms represents the most effective solution. Polytraumatized patients, however, might show injuries that prevent arm elevation or require the inclusion of shoulder girdle and arms in the field of view. In addition, arm positioning above the head would usually require an interruption after head CT and CT angiography of head and neck leading to an extended scan time, which would have to be weighed against the actual diagnostic benefit in a time-critical trauma setting. In clinical practice, reimaging after stabilization with elevation of the arms and may prove useful in this context, as stated in the limitations. Severe obesity causing artifacts poses a severe diagnostic challenge. Despite maximal exposure factors or automated tube current modulation across different areas of the body with varying thickness, considerable artifacts might nevertheless remain or even non-diagnostic scans might be caused due to obesity. Technical limits seem to have been reached for the time being in this context. In regard to breathing motion artifacts, which led to exclusion of two patients in our study, wake and compliant patients are prompted to hold their breath before the thoracoabdominal scan of whole-body trauma CT. Polytraumatized patients, however, are often clouded in their consciousness, sedated or intubated, hampering active cooperation during the scan. Respiratory triggering or endexpiratory breathing arrest controlled by a ventilator machine may prove useful in this context. Lastly, beam hardening artifacts due to radiopaque metallic foreign objects located in- and outside of the patient may impair image quality. Next to removal of non-essential radiopaque material prior to the CT scan or placement outside of the field of view, the use of modern metal artifact reduction techniques may prove useful in this context.

We thank Reviewer 1 for the remark and adjusted our M&M section as well Discussion section accordingly.

  1. You mention potential applications for automated quantification and AI-based detection. Could you expand on specific next steps or ongoing work in this direction?

So far, there is no ongoing work in this direction on our part. The background to our idea is to make use of the density values we have shown analogous to emphysema quantification using CT [4, 5].

  1. Given that your results are specific to one PCD-CT scanner model, how generalizable do you think these findings are to other spectral CT technologies?

The utilization of the enhanced iodine contrast is a basic principle of the VMIs and is manufacturer-independent. We therefore believe that our results are fully transferable to VMIs from other manufacturers. The extent to which the manufacturer-specific VMI+ algorithm for noise reduction in low-kev level VMIs applied in our study performs in comparison to other manufacturers or to what extent other manufacturers use similar algorithms, however, is beyond our knowledge and requires comparative studies. As a consequence, we included this aspect in our limitations.
We thank Reviewer 1 for this remark and consider this as a starting point for future investigations.

  1. The potential for VMI to improve existing methods of quantifying pulmonary injury is interesting. Do you have any preliminary data on correlations between VMI-based injury assessments and clinical outcomes?

We use this study as a starting point for further investigations in collaboration with the Department of Thoracic Surgery in our institution. Currently, there are no preliminary data available, but the proposed approach is undoubtedly promising and offers great potential for future exploration.

References

  1. Konietzke, P., et al., Consolidated lung on contrast-enhanced chest CT: the use of spectral-detector computed tomography parameters in differentiating atelectasis and pneumonia. Heliyon, 2021. 7(5): p. e07066.
  2. Edwards, R.M., et al., A Quantitative Approach to Distinguish Pneumonia From Atelectasis Using Computed Tomography Attenuation. Journal of Computer Assisted Tomography, 2016. 40(5): p. 746-751.
  3. Albrecht, M.H., et al., Comprehensive Comparison of Virtual Monoenergetic and Linearly Blended Reconstruction Techniques in Third-Generation Dual-Source Dual-Energy Computed Tomography Angiography of the Thorax and Abdomen. Investigative Radiology, 2016. 51(9).
  4. Ostridge, K. and T.M. Wilkinson, Present and future utility of computed tomography scanning in the assessment and management of COPD. Eur Respir J, 2016. 48(1): p. 216-28.
  5. den Harder, A.M., et al., Emphysema quantification using chest CT: influence of radiation dose reduction and reconstruction technique. Eur Radiol Exp, 2018. 2(1): p. 30.

Reviewer 2 Report

Comments and Suggestions for Authors

The paper explores the utility of virtual monoenergetic images (VMIs) generated from photon-counting detector CT (PCD-CT) in differentiating between severe lung injury and atelectasis in polytraumatized patients. The study concludes that this imaging technique can enhance diagnostic accuracy and improve decision-making in emergency situations.

According to the authors, the application of PCD-CT imaging and VMI reconstruction in the context of severe lung injury and atelectasis is innovative, offering significant clinical value. The study presents a well-structured methodology with clear inclusion and exclusion criteria, which strengthens its reliability. The paper presents a potential improvement in early diagnostics and treatment for patients with severe thoracic trauma, potentially reducing complications such as ARDS (acute respiratory distress syndrome).

However, the primary limitation of the paper is the relatively small sample size, which restricts the generalizability of the findings. Additionally, the authors do not clearly specify what was used as the standard of reference (e.g., follow-up studies) and how many of theses cases could not be discriminated by morphology solely. The exclusion of four patients (20% of the final population) raises concerns about whether this sophisticated reconstruction method is suitable for emergency settings, where artifacts are common.

Upon reviewing images 1 and 2, a question arises regarding how often the distinction between atelectasis and lung injury is impossible and critical for patient management in real-world settings. It is unclear whether the use of specific VMIs—requiring reconstruction at two energy levels—is necessary.

In trauma cases, timely image interpretation is crucial. Given the large volume of images that residents must review, combined with the need for multiple reconstructions, it is uncertain whether these two specific reconstructions significantly simplify the workflow. Considering the relatively low incidence of these patients and the greater importance of findings like fractures or pneumothorax, the added value of these VMIs may be limited.

Reviewing the image 1 and 2 the question rises, how often in real setting the discrimination between atelectasis and injury is an issue regarding patient management and needs the used of specific VMIs, which than have to be reconstructed on two energy levels. 

I assume that in trauma the reading time is crucial. If the resident already needs to review hunderts of images and a number of reconstructions have to be made I'm not sure if this two reconstruction are really making the work easier, as the number of these patients is very low and other findings like fractures or pneumothorax might be more important.

Minor point: Please put into the reference of Fig1, which ring corresponds to which pathology.

Author Response

Reviewer 2

The paper explores the utility of virtual monoenergetic images (VMIs) generated from photon-counting detector CT (PCD-CT) in differentiating between severe lung injury and atelectasis in polytraumatized patients. The study concludes that this imaging technique can enhance diagnostic accuracy and improve decision-making in emergency situations.

According to the authors, the application of PCD-CT imaging and VMI reconstruction in the context of severe lung injury and atelectasis is innovative, offering significant clinical value. The study presents a well-structured methodology with clear inclusion and exclusion criteria, which strengthens its reliability. The paper presents a potential improvement in early diagnostics and treatment for patients with severe thoracic trauma, potentially reducing complications such as ARDS (acute respiratory distress syndrome).

However, the primary limitation of the paper is the relatively small sample size, which restricts the generalizability of the findings. Additionally, the authors do not clearly specify what was used as the standard of reference (e.g., follow-up studies) and how many of theses cases could not be discriminated by morphology solely. The exclusion of four patients (20% of the final population) raises concerns about whether this sophisticated reconstruction method is suitable for emergency settings, where artifacts are common.

Upon reviewing images 1 and 2, a question arises regarding how often the distinction between atelectasis and lung injury is impossible and critical for patient management in real-world settings. It is unclear whether the use of specific VMIs—requiring reconstruction at two energy levels—is necessary.

In trauma cases, timely image interpretation is crucial. Given the large volume of images that residents must review, combined with the need for multiple reconstructions, it is uncertain whether these two specific reconstructions significantly simplify the workflow. Considering the relatively low incidence of these patients and the greater importance of findings like fractures or pneumothorax, the added value of these VMIs may be limited.

Reviewing the image 1 and 2 the question rises, how often in real setting the discrimination between atelectasis and injury is an issue regarding patient management and needs the used of specific VMIs, which than have to be reconstructed on two energy levels. 

I assume that in trauma the reading time is crucial. If the resident already needs to review hunderts of images and a number of reconstructions have to be made I'm not sure if this two reconstruction are really making the work easier, as the number of these patients is very low and other findings like fractures or pneumothorax might be more important.

We thank Reviewer 2 for her/his fair, constructive and helpful comments regarding our manuscript. Please find our responses to your remarks in the following.

In regard to a comparably small sample size included in our study, we agree that further studies are necessary to underline the generalizability of the results obtained, which are already underway. In the case of the descriptive proof-of-concept study design with statistically highly significant results, we consider our approach to be adequate and worth presenting, as the potential clinical benefit is clearly demonstrated and discussed.

As described in the Introduction section and concurring with our clinical experience, morphological criteria alone frequently hamper radiologists making a clear classification of trauma-associated lung opacification. It is not uncommon for both differential diagnoses to be listed as potentially underlying cause for opacification in the final report. Especially in the case of anatomically closely coexisting atelectasis and lung injury as well as lung injuries within atelectatic lung areas, VMIs proved to be a useful addition. In addition, the simultaneous occurrence of both entities in as many as 1/3 of all whole-body trauma photon-counting CT scans screened by us should be taken into account.

The potential clinical benefits coming with this novel application are extensively discussed in our discussion and are currently addressed by further studies on the clinical outcome initiated by us. As an example for benefits in clinical decision making in critical trauma setting, we firmly believe that clear identification of sole compression atelectasis without evidence for lung injury can be adequately addressed by insertion of a chest tube without the need for ICU treatment. On the other hand, exact identification of the extent of severe lung injury in uncertain situation with both compression atelectasis and lung injury leads to substantiated initiation of adequate therapy, such as ventilation therapy in an ICU. Both examples demonstrate the benefits of resource-oriented clinical decision making based on improved diagnostic imaging.

In regard to the workflow complexity for radiologists, we would like to provide further insights into our workflow. Standard reconstructions of whole-body trauma photon-counting CT are immediately followed by spectral reconstructions using spectral post-processing (SPP) datasets. The time interval between completion of the standard reconstructions and completion of the spectral reconstructions is 3 to 4 minutes. In practice, the spectral data are fully reconstructed when you are finished assessing the standard reconstructions. Using a manufacturer-specific workstation (Syngio.via, versionVB60A_HF04, Siemens Healthineers, Forchheim, Germany) an optional change of Monoenergetic Plus keV levels (40-200 keV) right within the application mode is available. This way, the advantages of both high-energy (e.g. metal artifact reduction) and low-energy virtual monoenergetic images (e.g. improved iodine contrast) can be exploited immediately in the emergency setting without substantial time delay. This way, an additional diagnostic tool is provided on demand for specific assessment without the need to assess the entire dataset.

It is undisputed, that artifact burden represents a major challenge in whole-body trauma CT. Further elaborations concerning this, including potential strategies to overcome this limitation, were added to our manuscript in this review process following a similar remark raised another Reviewer. Nevertheless, we were able to present diagnostic VMIs in the majority of patients included.

Minor point: Please put into the reference of Fig1, which ring corresponds to which pathology.

We revised the figure legend accordingly.

Reviewer 3 Report

Comments and Suggestions for Authors

Wonderful work by the authors. A couple questions to clarify.

1. In the introduction you mention using VMI to help discriminate several things including infection in trauma, but to my reading there is no and likely no infectious etiology in this cohort? 

2. These trauma CTs are presumably done rapidly. Can the authors clarify a time frame for when the scans are performed and whether this plays a role in the ability to discriminate trauma from atelectasis. 

3. Does the presence of an endotracheal tube or chest tube alter the ability for the image to be interpreted as effectively as what is reported? 

Great work

Comments on the Quality of English Language

Only minor spelling errors to note

Author Response

Reviewer 3

Wonderful work by the authors. A couple questions to clarify.

We thank Reviewer 3 for her/his valuable and fair remarks. Please find a point-by-point response in the following.

  1. In the introduction you mention using VMI to help discriminate several things including infection in trauma, but to my reading there is no and likely no infectious etiology in this cohort? 

That is correct. In our Introduction section, we listed pneumonia as a potential late complication of severe lung injury among several other complications (pleural fistula, persisting pneumothorax, abscess formation, pleural empyema or acute respiratory distress syndrome), in order to underline the necessity of clear and early identification of severe lung injury.

  1. These trauma CTs are presumably done rapidly. Can the authors clarify a time frame for when the scans are performed and whether this plays a role in the ability to discriminate trauma from atelectasis. 

Standard reconstructions of whole-body trauma photon-counting CT are immediately followed by spectral reconstructions using spectral post-processing (SPP) datasets. The time interval between completion of the standard reconstructions and completion of the spectral reconstructions is 3 to 4 minutes. In practice, the spectral data are fully reconstructed when you are finished assessing the standard reconstructions. Using a manufacturer-specific workstation (Syngio.via, versionVB60A_HF04, Siemens Healthineers, Forchheim, Germany) an optional change of Monoenergetic Plus keV levels (40-200 keV) right within the application mode is available. This way, the advantages of both high-energy (e.g. metal artifact reduction) and low-energy virtual monoenergetic images (e.g. improved iodine contrast) can be exploited immediately in the emergency setting without substantial time delay.

  1. Does the presence of an endotracheal tube or chest tube alter the ability for the image to be interpreted as effectively as what is reported? 

In our study, we did not encounter any relevant artifacts due to a chest tube or endotracheal tube. Beam hardening artifacts due to radiopaque objects were most times due metallic wires or sensors located outside of the patient. None of these artifacts, however, led to exclusion in our study.

Severe artifacts, that led to exclusion, were streak artifacts caused by photon starvation in non-elevated positioning of the arms aggravated by concomitant obesity and caused by breathing motion artifacts. Following a remark by Reviewer 1, we added further information concerning artifacts, including potential strategies to overcome this limitation, to the revised manuscript, which we kindly refer to.

Round 2

Reviewer 1 Report

Comments and Suggestions for Authors

Thank you for considering my suggested revisions for your manuscript. It is clear that you have put a lot of effort into refining of your work by integrating the feedback provided during the first round of the peer review process. The resulting changes have significantly improved the rigor and overall quality of your manuscript. I look forward to witnessing the impact your research will have on the clinical and academic community.